

# On the production and validation of satellite based UV index maps

Verena Schenzinger[1], Axel Kreuter[1,2], Barbara Klotz[1], Michael Schwarzmann[1], and Julian Gröbner[3]

[1]Institute for Biomedical Physics, Medical University Innsbruck, Innsbruck, Austria
[2]Luftblick OG, Innsbruck, Austria
[3]Physikalisch-Meteorologisches Observatorium Davos, World Radiation Center (PMOD/WRC), Davos Switzerland

**Correspondence:** Verena Schenzinger (verena.schenzinger@i-med.ac.at)

**Abstract.** This paper presents a method to calculate surface UV index maps from SEVIRI satellite images using a very simplified approach to account for clouds. We compare the resulting maps, which are produced near-real time operationally, to ground measurements from the Austrian UV network and affiliated stations. While the data agrees well for low altitude stations and in clear sky conditions, discrepancy at mountain stations and in certain cloud conditions between the map pixel and ground measurement UV indices gets higher. We discuss the sources of uncertainty in both values in detail, and highlight why a direct comparison of absolute UV index values for validating purposes is inadequate.

## 1 Introduction

About 30 years after the first definition of the UV index (Kerr et al., 1994) and its standardization (WMO, 1995), awareness and understanding of it in the general public is still low, especially in Europe (Heckman et al., 2019). Despite having lower exposure due to higher latitudes and generally higher stratospheric ozone levels, compared to e.g. Australia, the UV index still gets high enough to warrant protection. While the phenomenon of the "ozone hole" is mainly associated with the Southern Hemisphere, in April 2020 strong ozone depletion due to a split of the Arctic polar vortex was observed (CAMS, 2020), leading to higher UV exposures in adjacent regions (Alaska, Canada, Greenland, northern Europe and Russia). Events such as this might become more common with climate change, which emphasises the importance of continually monitoring UV radiation and informing the population about its health implications.

Following the Montreal Protocol in 1987, many countries established UV measurement stations. These ground based measurements are excellent for monitoring and providing local information, however, they usually cannot cover a large area, as local conditions of aerosol, ozone, clouds, or snow cover may vary, all of which influence the UV index. Satellites, on the other hand, can provide great area coverage. However, their imagery is not provided continuously, and they can only measure proxy data, but not UV index at ground per se. Nonetheless this data can be employed to produce UV maps (Verdebout, 2000; Schallhart et al., 2008; Kosmopoulos et al., 2021) effectively.

A lot of previous works distinguish cloudfree from cloudy situations by employing a radiative transfer model for clear sky calculations and a separate one to account for the cloud effects (Verdebout, 2000; Schallhart et al., 2008; Chubarova et al., 2012; Lakkala et al., 2020), but there are approaches to integrate all sky conditions in one algorithm (Kosmopoulos et al., 2021). In most models, clouds are parametrised as a homogeneous layer of constant thickness with varying liquid water content from





which the modification of the UV radiation is derived. (Verdebout, 2000) and (Schallhart et al., 2008), for example, use two lookup tables additionally to the clear sky lookup table in their surface UV index calculation: one to estimate the liquid water content of the cloud, and a second one to calculate the corresponding change in UV. We suggest an even more simplistic approach allowing us to derive the modification of the clear sky UV index due to clouds from a linear interpolation of irradiance measured at the satellite.

The quality of maps or other satellite based surface UV products is usually assessed by looking at absolute and relative deviations from ground data, which is basically all there is available to serve as the "ground truth". However, there are fundamental differences in these two datasets which might limit the accuracy of such a validation. In this paper we will discuss the problems of the satellite-ground comparison in detail using examples of the Austrian UV network.

## 2   Production of surface UV index maps

The UV index is a dimensionless number defined as the integral over the solar spectral irradiance $E(\lambda)$ multiplied with the erythemal action spectrum $s(\lambda)$ with respect to the wavelength $\lambda$:

$$UVI = k \int_{\lambda_0}^{\lambda_1} E(\lambda) \cdot s(\lambda) \mathrm{d}\lambda \tag{1}$$

where $\lambda_0 = 250\mathrm{nm}$, $\lambda_1 = 400\mathrm{nm}$ and constant $k = 40\mathrm{m}^2\mathrm{W}^{-1}$ (WHO/WMO, 2002).

It is important to note that the solar spectral irradiance refers to the one at ground level, not top of the atmosphere, which means that it is dependent on atmospheric composition, especially the ozone concentration, but also aerosol optical depth.

Clouds obviously modify the irradiance at the surface; while they usually reduce the irradiance at the surface, special conditions can also lead to UV enhancement compared to the expectations of a clear sky (Pfister et al., 2003; Calbó et al., 2005). The UV maps calculated for the Austrian UV network (Schallhart et al., 2008) are obtained by separating the atmospheric from the cloud effects as

$$UVI = \mathrm{CMF} \cdot UVI_{clearsky} \tag{2}$$

where $UVI_{clearsky}$ is calculated with the radiative transfer model libRadtran (Mayer and Kylling, 2005; Emde et al., 2016). The cloud modification factor (CMF) is calculated from satellite images at 600nm recorded by the SEVIRI instrument (EUMETSAT, 2017), which is the shortest available wavelength. We only do this calculation if the respective pixel is considered cloudy by the cloudmask obtained from the MSG Meteorological Products Extraction Facility (EUMETSAT, 2015). Both satellite products are resampled onto a regular lat-lon grid using nearest neighbour resampling as implemented in the pyresample package of the SatPy library.

To limit computation time, we use a precalculated lookup table for the clear sky UV index. The grid points (see table 1) were chosen so that the discrepancy between an interpolated and calculated value is less than 1%. The eccentricity of the Earth is taken into account by a separate multiplication factor as found in the libRadtran source code. Input for the clear sky model is





provided by the Copernicus Atmospheric Monitoring Service (CAMS) global model forecast (CAMS, 2021) and the digital elevation model GTOPO30 (Gesch et al., 1999). We use the standard aerosol settings in libRadtran according to (Shettle, 1989) and further parametrise the aerosol optical depth $\tau$ with the Angstrom exponent $\alpha = 1.4$ and variable $\beta$ following the relation $\tau = \beta\lambda^{-\alpha}$ (Ångström, 1929). We use CAMS forecast data in general as the algorithm is designed to produce near-real time

maps, but for retrospective analysis employing the best available input data should be considered.

We parametrise the cloud influence in a very simplified manner, which is essentially a min-max scaling of the irradiance at the top of the atmosphere: we equate a thick cloud (CMF = 0) with an albedo of 1 and therefore maximum radiance ($L_{max}$) at the satellite, and the absence of clouds (CMF = 1) with the actual ground albedo and the minimum radiance ($L_{min}$). Both the minimum and maximum radiance at the top of the atmosphere are calculated with a second lookup table which takes into account

geometrical factors (distance Sun-Earth, solar zenith and azimuth angle, satellite viewing angle) as well as the (Lambertian) surface albedo. With these two boundary values, the CMF is calculated from the linear interpolation between the two at the point of the measured radiance in the 600nm channel, $L_{\mathrm{VIS006}}$ (see figure 1):

$$\mathrm{CMF} = 1 - \frac{L_{\mathrm{VIS006}} - L_{min}}{L_{max} - L_{min}} \tag{3}$$

While this is an extremely simplified cloud model, figure 2 shows that it is not unreasonable: here, we used libRadtran to

simulate a homogeneous cloud in the layer between 2km and 3km height with liquid water content varying from 0.0 to 1.5 $\mathrm{gm}^{-3}$ and recorded the 600nm radiance at the top of the atmosphere as well as the 300nm radiance at ground level for a solar zenith angle of $45°$. To be able to compare the results for different albedos, we apply a min-max-scaling to the respective 600nm radiances and divide the 300nm radiance by the value in clear sky conditions (i.e. liquid water content of the cloud equals zero), which equals the CMF. For ground albedo up to 0.5, the relation of the two values can be reasonably approximated as linear.

For the ground measurements, we use broadband filter radiometers, either the Solar Light Model 501 (SL501), or a UV-S-E-T from Scintec – Kipp & Zonen. The measurements are averaged over 30 minutes for both detector types. Additionally to the 13 stations in Austria, we also include data from partner stations of the Austrian UV network at Davos and Weissfluhjoch (Switzerland), Leifers and Ritten (Italy), as well as Zugspitze and Munich (Germany) (Schallhart et al., 2008). We use quality controlled data from 2020 as this is the last year when measurements from the Hafelekar station are available.

## 2.1 Results

Figure 3 shows the two input files from the satellite, as well as the UV index map calculated with the clear sky model and the final UV index map which is the clear sky multiplied by the CMF. We show Austria and some of the neighbouring areas, including an indication for the location of our measurement stations, as an example. For this area (0.05° resolution, $180 \times 70$ pixels) the calculation for clear sky and real time map takes about 2 minutes on an office computer. However, the method per se

is not limited to a specific location in its application as long as the necessary input data is available. We operationally produce near real time UV index maps for Europe every 15 minutes (which is the measurement interval of the MSG satellite).

Figure 4 shows the comparison of the UV index measured at ground and the UV index of the map pixel of the respective





measurement station. Clear sky UV indices lower than one have been excluded for better visibility, and a distinction between cloudy and clear conditions has been made according to the value of the cloudmask.

### 2.1.1 Clear sky conditions

Performing a linear fit with a set intercept of 0 to the ground and map UV index in clear sky conditions yields slopes $r$ of $1 \pm 0.1$ for most stations. Exceptions are Bad Voeslau ($r =1.204$), Innsbruck ($r =1.103$), Leifers ($r =1.147$), Munich ($r =1.114$) and Zugspitze ($r =0.887$). The coefficient of determination $r^2$ is $> 0.9$ for all of the stations apart from Hafelekar ($r^2 =0.82$), which indicates a very good fit (for a perfect fit, $r^2 = 1$). Separating the stations into mountain ($> 1000$m a.s.l.) and low stations shows better agreement between ground and map UV index for the latter, with $r^2 > 0.95$ for all of them. For the mountain stations, $r^2$ can range from 0.82 (Hafelekar) to 0.97 (Davos).

For most stations the clear sky model has a positive bias, i.e. the satellite map overestimates the ground UV index measurements, quantified by the slope $m$ of the linear fit being larger than 1. Exceptions are the mountain stations at Ritten ($m =0.987$), Weissfluhjoch ($m =0.925$), and Zugspitze ($m =0.887$), where the satellite slightly underestimates the ground.

### 2.1.2 Cloudy conditions

In cloudy conditions, the agreement between map and ground UV index becomes generally worse. Using the same linear model as for cloudfree conditions, which incorporates possible biases of the clearsky model, $r^2$ drops below 0.9 for all stations, and as low as 0.55 for Sonnblick. Other stations for which the fit quality declines strongly are also mountain stations: Hafelekar (0.67), Weissfluhjoch (0.59), and Zugspitze (0.57).

For a further assessment of the cloud model itself, we compare the CMF values instead of the UV index: Figure 5 quantifies the agreement of map and ground CMFs in all sky conditions: it shows the correlation of the two values (Pearson's $r$) as well as the slope and intercept of a linear least squares fit. For all stations below 2000m a.s.l. the correlation is above 0.8, and the offset between the two datasets is close to 0 for all stations below 1000m a.s.l..

These results indicate a good agreement beween the ground and the map UV index in general, albeit declining with the station elevation.

## 2.2 Discussion

When evaluating solar UV index maps it is very common to compare the value of the ground measurement with the value of the map pixel that station is located in (this paper is no exception per se). The typical metrics assess the difference of these two values, or the related daily dose rate, in terms of their mean bias and root mean square error (Tanskanen et al., 2007; Lakkala et al., 2020; Kosmopoulos et al., 2021, e.g.). Apart from not addressing the underlying assumption of a random Gaussian error when using these metrics, this approach has other flaws which we would like to discuss in the following.



### 2.2.1 Distribution of the UV index

As can be seen from figure 4 or (Kosmopoulos et al., 2021, figure 5), the distribution of UV index values is skewed towards lower values due to the daily cycle and cloudy weather. Values of 9 or higher are only surpassed at a handful of stations; in Austria only in the high mountain stations, in Europe in general also by more Southerly locations. This will also influence the distribution of the differences, as large differences are only possible if the clear sky UV index is high enough itself.

Figure 6 illustrates this problem further. We show the map-ground difference in UV index, using the CMF calculated as described above and a "poor man's CMF" ($CMF_{0.5}$) where we set CMF = 0.5 for cloudy points and CMF = 1 otherwise. Furthermore, we draw a random CMF for cloudy points from a lognormal distribution capped at 1, also set to CMF = 1 for clear points.

While the $CMF_{0.5}$ (and random CMF) distribution is broader and does not peak as high as the modelled CMF distribution, there are still 57.5% (53.5%) of the points within a difference of $\pm0.5$ and 78.3% (74.0%) within $\pm1$, which would be classed as low to moderate difference by (Kosmopoulos et al., 2021). In comparison, the modelled CMF has 68.0% of points showing a difference of less than 0.5 and 86.8% of less than 1. When limiting the values to clear sky UV index $>1$, these percentages change to 55.0%/76.9% ($CMF_{0.5}$), 51.1%/72.3% (random CMF) and 66.1%/86.0% (modelled CMF) respectively, which lowers the apparent quality a bit, but $CMF_{0.5}$, and even the random CMF, still look quite respectable. This means that the approach of validation with histograms and simple statistics is not suitable for judging the quality of an all sky UV index model.

An alternative suggestion would be to compare the CMF directly to the ratio of the ground measurement to the clear sky model (which is, comparing with equation 2, effectively a ground CMF). Figure 7 shows the respective distributions of CMF differences from which the (expected) flaws of $CMF_{0.5}$ and the random CMF are quite obvious. Despite not all studies using the separation of UV index into a product of clear sky and CMF (Kosmopoulos et al., 2021), the ratio of the modelled UV index to a clearsky model can still be used in the comparison, which has a less skewed distribution compared to the UV index itself.

### 2.2.2 Differences between satellite and ground data

When validating satellite products, ground based measurements seem like a good point of reference. However, when it comes to the UV index, several fundamental differences could mean that this is a comparison of two different, yet identically named, parameters.

Broadly speaking, the ground measurement is taken at one point in space, often averaged over time, whereas the satellite represents one point in time averaged over a pixel (which, in the case of Austria, is about $4 \times 7$km in size).

For clear sky conditions, the satellite measurement, which is taken every 15 minutes, can differ by up to 0.5 UV index from the 30 minute average of the ground station, depending on when it is taken relative to the middle of the interval. This can be estimated by comparing values for the clear sky model every minute with respective 30 minute averages. For rapidly changing cloudy conditions, the difference between ground and satellite UV index value can become exacerbated, depending on the exact moment of taking the satellite image. The pixel averaging means that small scale or inhomogenous clouds cannot be resolved by the satellite, leading to poor skill in detection of broken cloud conditions as well (Werkmeister et al., 2015).





Figure 8 shows a situation where that timing worked quite well and the scale of the clouds was large enough so that the station was representative for the whole pixel: after a sunny first half of the day, rain clouds developed in Western Austria (satellite image at 14 UTC is shown in figure 3), leading to a drop in the UV index. For the stations at Innsbruck and Kirchbichl (located about 70km further down the Inn valley), the satellite captures this change well in terms of timing and magnitude. On the other hand, figure 9 illustrates some limits of the satellite based modelled CMF: this day was a quite typical summer day with

cumulus clouds developing over mountain peaks, while the valley stayed sunny. However, these clouds are very localised, which means that despite the Hafelekar peak (which is only about 7km linear distance away from Innsbruck, i.e. one pixel for the satellite) being covered in clouds, the satellite saw the corresponding pixel as cloudfree, leading to a discrepancy between ground and modelled UV index. For Innsbruck, the model would still be considered good despite a small bias in the clear sky model and not resolving the thin clouds during midday (mean difference is 0.4 UV index, 77% of points have less than 0.5

difference).

Even when calculating the CMF for the Hafelekar pixel regardless of what the cloudmask was identifying it as, its values would still be equal to 1 most of the time, or at least >0.98, i.e. basically the same as if there were no clouds at all. In these particular meteorological conditions, representativeness of a station for a pixel gets lowered (Kazadzis et al., 2009; Slobodda et al., 2015; Lakkala et al., 2020) and cloud observations from the ground and the satellite are fundamentally different due

to differences in recording and viewing geometry (Malberg, 1973; Carbajal Henken et al., 2011; Bieliński, 2020). This is the case in particular for small scale clouds like convective cumuli over mountain tops, or broken cloud conditions. Nonetheless, both the satellite, recording the average over the pixel, and the ground detector, influenced by the local conditions, are correct from their respective point of view. However, the two UV index values cannot be compared to each other in a meaningful way, let alone be used in the context of the ground measurement validating the satellite data (figure 9). For large scale clouds, like

fronts or generally overcast sky, the comparability is much better and the modelled UV index is in accordance with the ground (figure 8).

### 2.2.3  Uncertainties in the ground CMF

Defining the ground CMF as the ratio of measured to modelled UV index means that there are two sources of uncertainty: the measurement error itself, and the clear sky UV index model.

Uncertainties in the ground measurement are due to calibration uncertainties (Blumthaler, 2004; Hülsen et al., 2008) and changes in the detector in between calibrations which can be as high as ±10% (Schallhart et al., 2008). Intercomparisons of broadband radiometers have shown great variability of instruments with up to ±50% difference compared to a reference instrument (Gröbner et al., 2006), though the type of instruments used by the Austrian UV network perform way better at less than ±15% deviation.

The accuracy of the clear sky model depends mainly on the quality of its input data. Figure 10 shows how different inputs can influence the UV index, using 21 March midday at the Hafelekar station as an example. For this particular date, an error of 0.025 in $\beta$ leads to a UV index difference of 0.15, an error of 0.1 in albedo to $\Delta UVI = 0.2$. This means, that 10% variation in either of these two parameters lead to roughly 3% variation in UV index. For ozone, which is the most influential parameter,



but also usually the least error affected of the three, the effect on UV varies depending on the base level. As a rule of thumb,
a change of 1% in Ozone concentration leads to a change of ∼1% in surface UV radiation. Typically, ozone levels in Europe
are around 300-400 DU and the error in CAMS is usually less than 10 DU (Inness et al., 2019), which would mean ∼ ±5%
difference in UV index.

The morning of the day shown in figure 8 exemplifies the uncertainties of the input data: the ratio of the measurement to
the clearsky model is below 0.9, and therefore outside the instrument error. The CAMS forecast value for ozone was 315
DU, whereas the reanalysis, which assimilates measurements, was 305 DU; the local Pandora instrument measured 308 DU
for midday. However, this means that the clearsky model used was actually too low by about 3%. This underestimation is
counteracted by using a (broadband) albedo of 0.17, whereas an UV albedo below 0.1 is more realistic for summer. That
leaves an erroneous input for aerosol as the source of error. Measurements from the local sun photometer and the Sonnblick
observatory (Schauer, 2021) confirm a Saharan dust event. In this case the Angstrom relation is not appropriate to derive the
aerosol optical depth, which was effectively higher than estimated, meaning the UV index was overestimated in the clear sky
model.

These two sources of error explain the differences between map and ground UV index (figure 4) for clear sky conditions as the
map value is identical to the clear sky model (as CMF = 1), at least for the stations below 1000m. For others, snow cover means
that the error in albedo can be much larger as the suggested 10%, because snow free ground has very low UV albedo (less than
0.1), whereas for fresh snow it can be as high as 0.9. Furthermore, snow cover is very localised, so averaging over a pixel leads
to underestimation of albedo at the peaks and overestimation in the valley. We discuss this further in the next section.

### 2.2.4 Uncertainties in the map CMF

Before satellite data is used for map production, it is usually reprojected onto a regular latitude-longitude grid. This can be done
in different ways; the pyresample module offers three different options for gridded data: nearest neighbour, bucket averaging,
bilinear interpolation (Pyresample developers, 2013–2023). While it is not sensible for the cloud mask, which is basically
binary data, to use interpolated values, the 600nm irradiance can be processed with any of these algorithms. Figure 11 shows
how the method can affect the CMF calculation, comparing the values obtained for Austria during one day. The difference
between the CMFs obtained with the nearest neighbour vs. the bilinear interpolation method is usually below 0.05 - this means
for low CMFs it can be up to a 50% difference, whereas above CMF = 0.5 the difference gets less than ∼ ±10%, which is
in the order of the ground measurement uncertainty. Similarly, it matters how the regular latitude-longitude grid is defined: a
diagonal shift of half a pixel (i.e. moving the middle to the lower left edge) can easily lead to a deviation of ∼ ±10% in the
resulting CMF. Both these choices add to the error of cloud affected points in the comparison of ground and map values (figure
4), potentially leading to a higher variation in difference for cloudy compared to clear sky conditions.

Albedo is the only parameter that leads to uncertainty in both the clear sky model and the CMF calculation, leading to a two-
fold error: for the clear sky model, an overestimation also increases the UV index (see figure 10). Simultaneously, erroneously
high albedo means that the interval between minimum and maximum irradiance shrinks, leading to a higher CMF (equation
3) and higher uncertainty, as little variation in the irradiance can lead to high differences in CMF (figure 12). This means that



not only is the clear sky UV index too high, but also the CMF, giving an overestimation of the overall ground UV index, which is often the case for mountain stations (figure 4). Even if the cloud mask, which uses a quite sophisticated cloud detection
algorithm, does correctly identify a pixel as cloudy, the CMF calculation will still be highly erroneous if the ground albedo is high.

Figure 13 illustrates the problem hinted at in the previous section: the stations of Hafelekar and Innsbruck are in the same pixel in the CAMS global forecast, which has a resolution of $0.4°$. This means that in winter, the ground albedo at Hafelekar (2269m a.s.l.) will generally be underestimated, whereas at Innsbruck (578m a.s.l.) it will be overestimated. For the satellite CMF,
this gives an underestimation for Hafelekar and an overestimation for Innsbruck and vice versa for the ground CMF, resulting in most high albedo points being below the equivalence line for Hafelekar, and above for Innsbruck. This figure furthermore shows that the variability in ground vs. map CMF is higher for the mountain station than for the valley, which means that the local conditions lead to the station not being representative for a larger area.

## 3 Conclusions

We present a computationally cheap method to calculate ground UV index near real time maps from a combination of atmospheric model forecasts and satellite images (figure 3). This is done by a separation of the UV index into a clear sky model and the cloud contribution in the form of a modification factor (CMF) (equation 3). This approach itself is widely employed (Verdebout, 2000; Schallhart et al., 2008; Chubarova et al., 2012; Lakkala et al., 2020), because clouds introduce the highest variability in UV index (He et al., 2013), whereas the clear sky UV index can be predicted very well with radiative tranfer
models if information on ozone, aerosol, and surface albedo is available in sufficient quality. What makes our method different is the highly simplistic, yet reasonable treatment of parametrising the clouds (figures 1 and 2).

We compare the results to measurements of stations in the Austrian UV network and associated partners. For clear sky conditions and low altitude stations, the agreement of the UV index map pixel with the ground measurement is within the 10% range of ground measurement uncertainty. This is true for about 90% of the higher altitude stations as well. For cloudy conditions,
the deviation of map to ground is within ±10% for 60-70% of the low stations, but can drop to ∼30% for the mountain stations. The seemingly bad agreement for the mountain stations is due to them having very specific local conditions (albedo, aerosol, cloud) and therefore not being representative for the whole satellite pixel.

We point out that the comparison of UV indices is flawed due to the underlying skewed distribution of possible values, which can get slightly alleviated by excluding clear sky values below UV index 1. In public communication and for health purposes,
this is still the right parameter to choose, as it is the one relevant in the context. However, for the purpose of model validation, the CMF (or the mathematically identical ratio of UV index to a clear sky model), is the better choice. Nonetheless, the simple approach of looking at histograms of absolute UV index differences is not suitable for judging the quality of a UV model, as a very simplified CMF model of a constant CMF or even a random CMF still perform well by this standard (figure 7).

Further, we discuss the differences in the ground measurements and the maps in detail, including sources of uncertainties in
both. Particularly in the presence of small scale or broken clouds, the map UV index and the ground UV index become basi-





cally incomparable. Taking the cloud type into account during production of the satellite maps means that those could not only show the UV index value itself, but also its reliability/accuracy for the respective pixel. How to derive this type of information, ideally from the input already used in the map production, and to convert it into an accuracy measure, is subject to further study.

*Data availability.* The measurements of the Austrian UV network are available on request as of the publishing date (public API access is planned). Recent UV index timeseries and current maps of Europe can be viewed at uv-index.at.

*Author contributions.* VS designed and implemented the CMF algorithm. MS calculated the clearsky lookup table. BK provided the UV index measurements and was responsible for quality assurance. The manuscript and figures were drafted by VS, with all authors contributing to its final state.

*Competing interests.* The authors declare no competing interests.

*Acknowledgements.* The authors thank Mario Blumthaler for valuable discussions.

The Austrian UV network is supported by a grant from the Federal Ministry for Climate Action, Environment, Energy, Mobility, Innovation and Technology (BMK).

This publication uses Copernicus Atmosphere Monitoring Service Information [2023]; neither the European Commission nor ECMWF is
responsible for any use that may be made of the information it contains.





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

**Figure 1.** Calculation of the CMF as a linear interpolation between minimum and maximum 600nm radiance at the satellite.





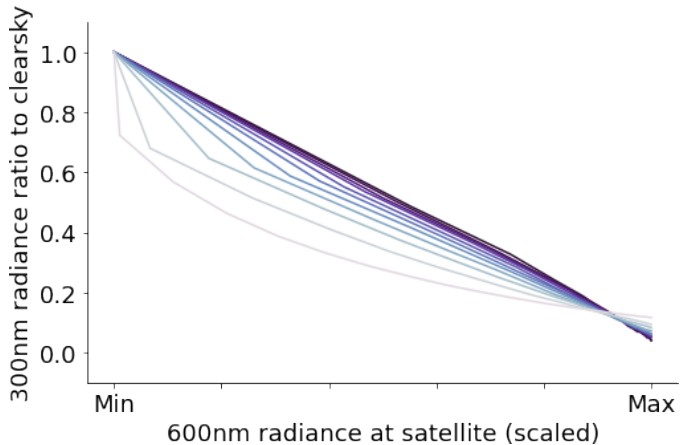

**Figure 2.** Ratio of the 300nm radiance at surface to cloudfree conditions versus the corresponding min-max scaled 600nm radiance at the top of the atmosphere (satellite measurement) for ground albedo 0 (dark) to 1 (light) as simulated with libRadtran.



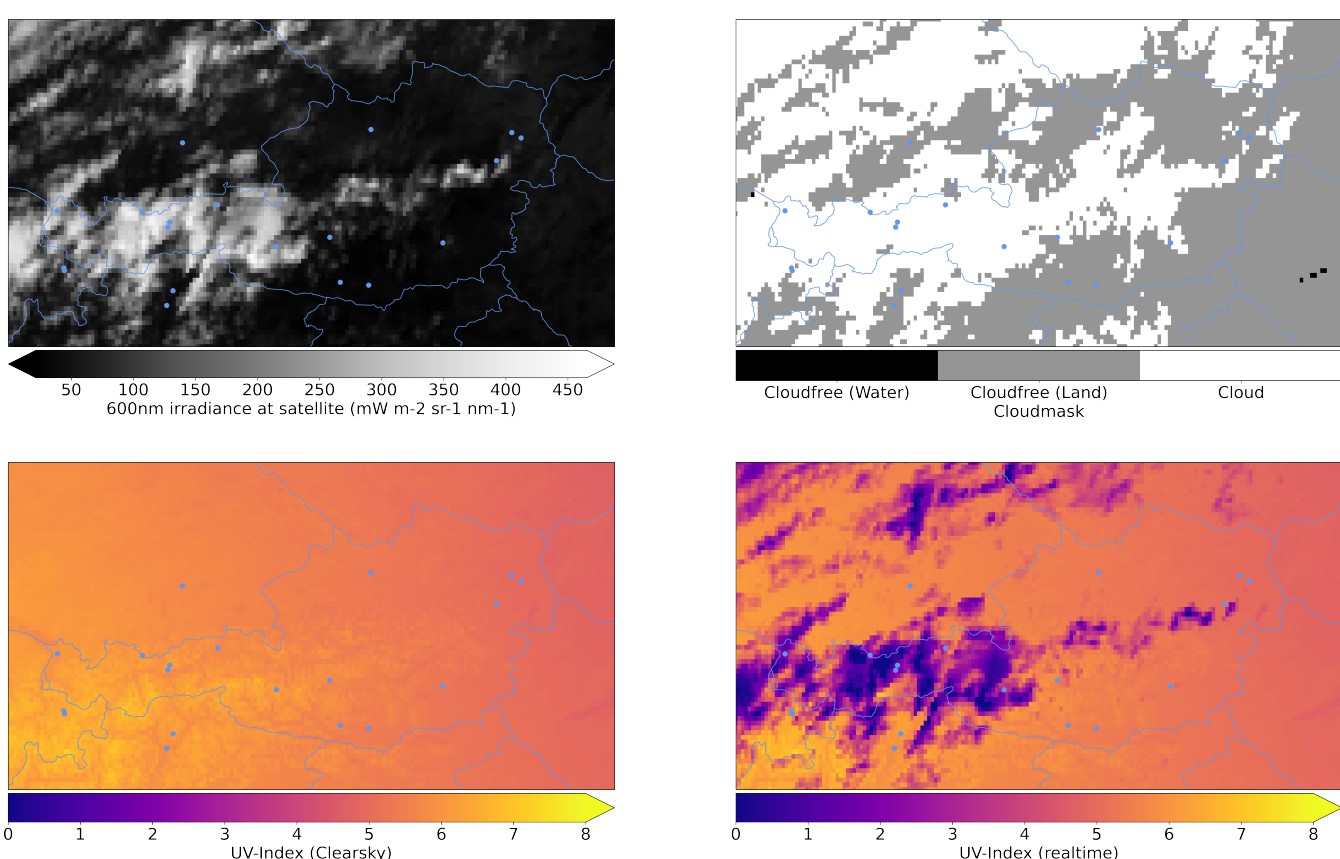

**Figure 3.** Input data from the satellite (top row), clear sky UV index map and real time UV index map taking clouds into account (bottom row).



**Figure 4.** Scatter plots of 2020 ground UV index vs. satellite map UV index for the stations in the Austrian UV network, sorted alphabetically. One/Two asterisks indicate that the station is higher than 1000/2000m a.s.l.. Data is limited to clear sky UV index higher than 1. Cloudy points according to the cloudmask are in red, clear points in blue.



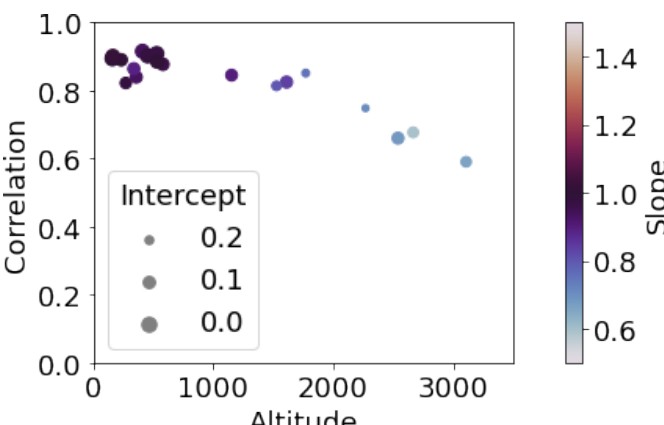

**Figure 5.** Correlation of the map and the ground CMF depending on the height of the measurement station. The colour indicates the slope of a linear least squares fit (ideal: 1) and the size the intercept (ideal: 0).



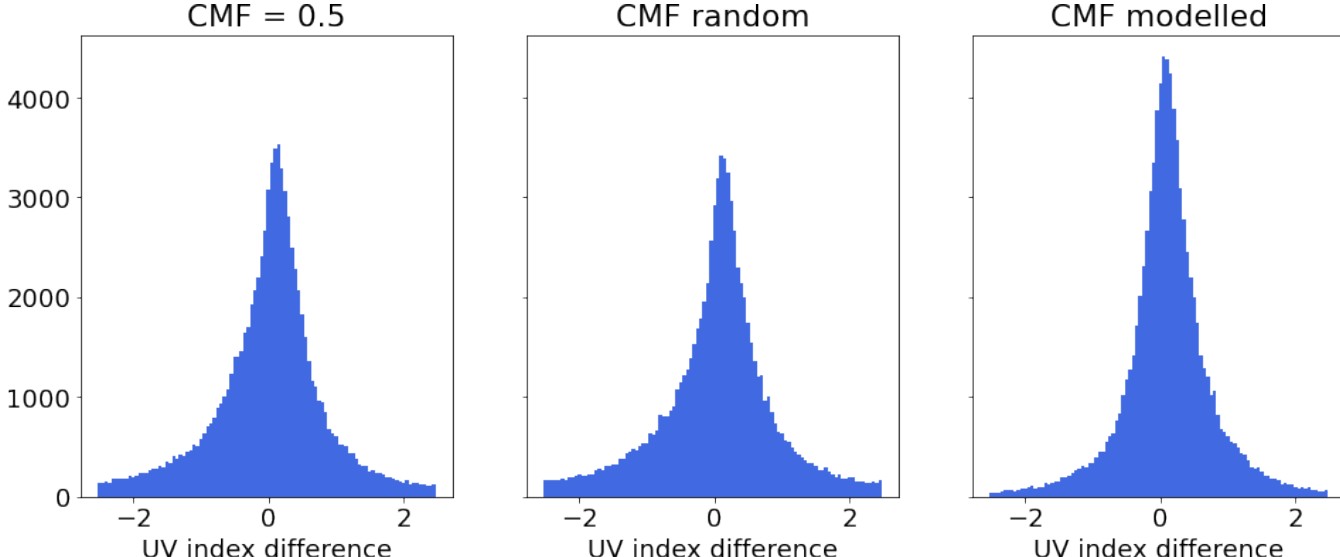

**Figure 6.** Histograms of map - ground UV index for all stations and sky conditions for (left) a constant CMF = 0.5 in cloudy condictions, (middle) a randomly sampled CMF, and (right) the modelled CMF calculated from satellite data.



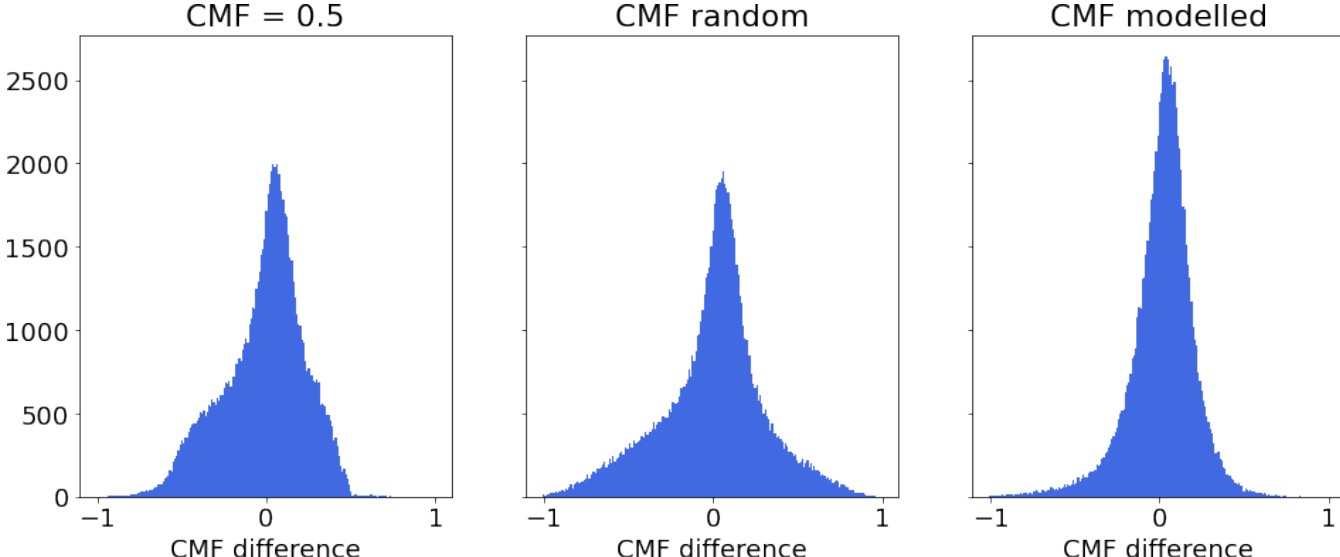

**Figure 7.** Histograms of map - ground $CMF_{0.5}$ (left), random CMF (middle), and modelled CMF (right) for all stations and sky conditions, limited to clear sky UV index > 1.

**Figure 8.** Top: Sky camera pictures (North is on top) taken in Innsbruck at 8, 10, 12, 14, 16 UTC on 2020/07/10. Middle: clear sky UV index (grey), modelled UV index (magenta) and ground measured UV index (green). Bottom: Ratio of UV index to clear sky model (equivalent to the CMF). Crosses indicate when the point is considered as cloudy.

**Figure 9.** Same as figure 8, but for 2020/07/20. The Hafelekar peak is part of the mountain range in the North of the camera pictures.





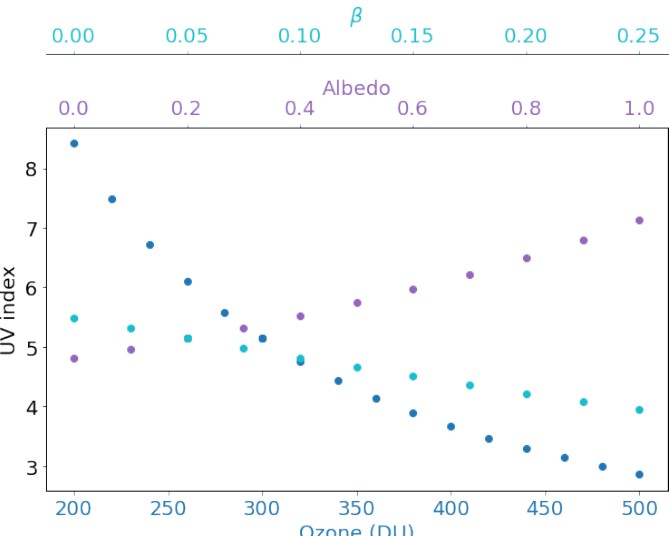

**Figure 10.** Clear sky UV index for Hafelekar station for March 21 depending on the model input parameters.

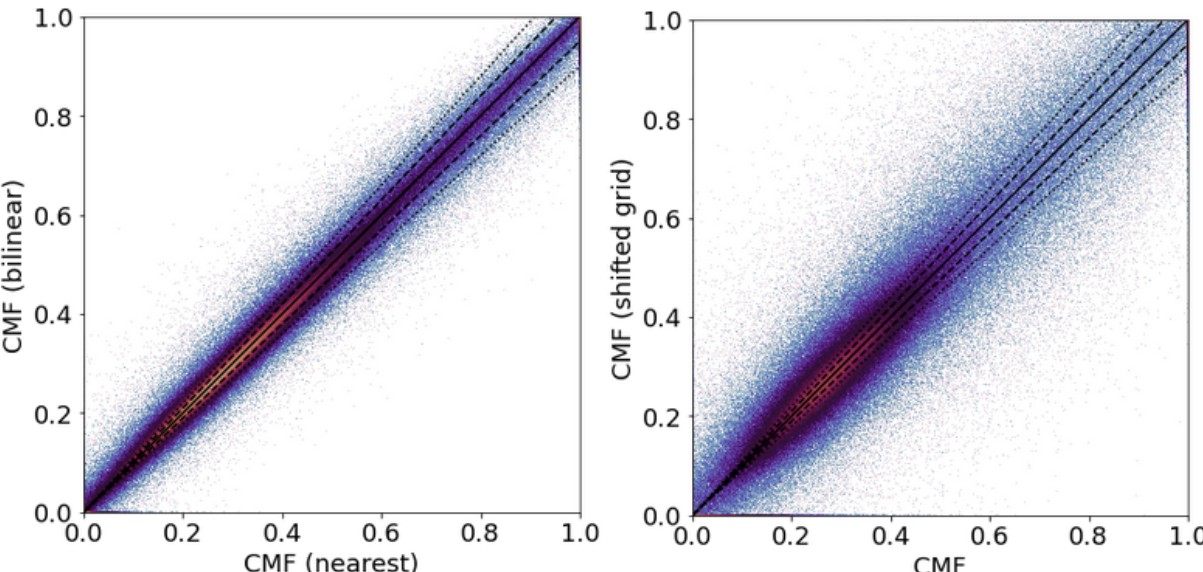

**Figure 11.** Left: Modelled CMF depending on the method used for resampling with pyresample: nearest neighbour (as used in this paper) vs. bilinear interpolation. The black line indicates equivalence, the dashed/dotted lines $\pm5\%/\pm10\%$ respectively. Right: Modelled CMF for a regular latitude-longitude grid vs. the same grid shifted by half a pixel. Colour indicates the density of the datapoints.



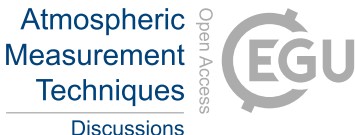

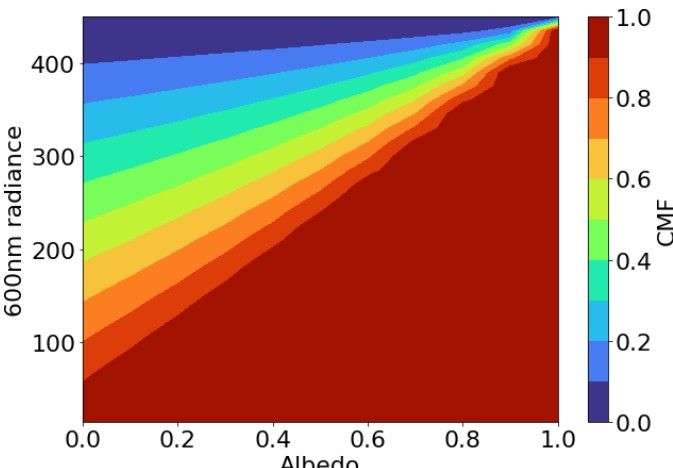

**Figure 12.** Modelled CMF depending on measured satellite irradiance and ground albedo.

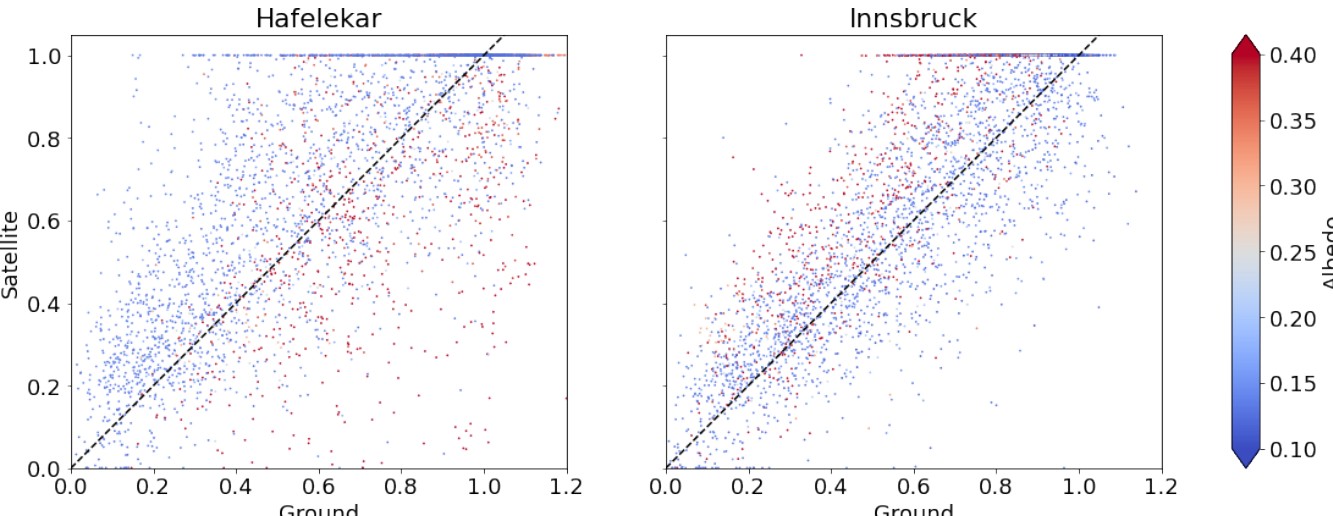

**Figure 13.** Scatter plots of ground CMF vs modelled satellite CMF for Hafelekar and Innsbruck. Colour refers to the albedo input from CAMS. The dashed line indicates equivalence of the two parameters. Note that the CMF values of the satellite model are capped at 1 (cloudfree pixels) by design, leading to the apparent horizontal line.



| Parameter | Unit | Values |
|-----------|------|--------|
| Albedo | 1 | 0, 0.12, 0.24, 0.34, 0.45, 0.54, 0.53, 0.71, 0.79, 0.87, 0.93, 1.0 |
| Ozone | DU | 160, 174, 188, 204, 221, 240, 260, 282, 306, 332, 360, 390, 423, 459, 498, 540 |
| Height a.s.l. | km | -0.5, 0.5, 1.5, 2.5, 3.5 |
| Solar elevation | ° | 42 values from -2 to 70 |
| Angstrom $\beta$ | 1 | 0, 0.01, 0.025, 0.045, 0.065, 0.09, 0.13, 0.17, 0.21, 0.25 |

**Table 1.** Grid points of the clearsky lookup table calculated with libRadtran's sdisort model.