# Peer review of "On the production and validation of satellite based UV index maps"

_Atmospheric Measurement Techniques, 2023_

## Author Comment (AC1)

The current manuscript presents a method to calculate surface UV index from SEVIRI satellite images at 600 nm using a simple linear method to account for cloud effects. The second part of the manuscript discussed the uncertainties in the comparison between modelled and measured UV index which are very useful to understand the difficulties of validation of satellite derived UV index. The manuscript is well written. However, I have some concerns about the proposed method. I would suggest major modification before the manuscript is able to be published.

**Major points:**

L28-30 & L235-236: One major point is why proposing such a simplified method? What is the advantage/disadvantage of this method comparing to the two lookup tables approach? Dose the gain of calculation time or other factors outweighs the lost of accuracy? More justifications or defined objectives are needed here.

*Compared to the previously employed method of using 2 lookup tables (one to estimate liquid water content and one to calculate the CMF), this simplified one is computationally very cheap. Furthermore, the remaining "expensive" operations (interpolating the minimum and maximum radiance map), can be done in advance, basically as soon as the albedo information is obtained. This means that at runtime there is only a subtraction and a division to calculate – this method can be run on a standard office PC and does not require any high power computing facilities (like e.g. the UVIOS project uses).*
*Given the uncertainty of the input parameters (like the albedo, especially with its variation on small scales like in the Alps, or the proper cloud parameters), the loss of accuracy due to the linear interpolation compared to the double lookup table, is indeed negligible.*
*We will add a paragraph to the introduction to make the motivation more clear and also exemplify the uncertainties with the experiments of a constant and random CMF.*

L69-72: For the results presented in Figure 2, the influences of surface albedo and liquid water content are examined. But many other factors are not clear here, e.g. the variation of cloud altitude and thickness, cloud ice water content for high clouds, solar zenith angle as well as aerosol properties. I believe that there should be some investigations about those factors to support the use of this linear method or to be clearer about the deficiency of using the method.

*We will change figure 2 to this unscaled version to show that a linear interpolation between the start and end point of each separate line is justified.*

[Figure]

*The details mentioned by the referee regarding the structure of clouds which could impact the UV radiation are all relevant. However, evaluating these parameters is associated with high uncertainties itself (see e.g. https://acp.copernicus.org/articles/22/8259/2022/) and even if it was not, the appropriate modelling of the clouds, which would have to be a 3D model, is computationally way too expensive for the purpose of real-time UV maps.*

**Minor points:**

End of the section 1: Maybe add a conventional paragraph to describe the structure of this manuscript?

*The authors believe that this kind of paragraph does not add any information to the manuscript, which follows the conventional structure.*

L97-98: Any possible explanations for this overestimation? Is this caused by the errors of aerosol amount or ozone?

*It is unlikely to be due to error in ozone, as there should be no systematic error in this variable. It is possible to be due to aerosol (if you compare the stations Innsbruck and Kirchbichl, which are climatologically very similar, but Innsbruck is more aerosol affected, the bias is less in Kirchbichl). Another issue could be that topography on the horizon is blocking part of the diffuse UV radiation.*

L99: Change "the ground" to "the ground UV" at the end of the phrase?

*Ok.*

L168-169: I'm not sure about this argument. The goal of having satellite based UV index is to provide estimations of what will be observe at ground level (although they include errors too). If satellite UV index is different from the ground observation, then it should be that the satellite UV index has deficiencies, either due to the modeling methods or the limits of satellite instrument itself, e.g. SEVIRI's capability to capture small scale cumulus clouds.

*We will try to make the point more clear. There definitely are errors in the ground measurements as well, so these are not the absolute "ground truth". The main issue however is, that differences between the ground value and the map value are not necessarily due to errors (of one value or the other), but can also be caused by situations in which the (point) measurement of the ground is not representative for the map pixel.*

L198-200: The proposed method itself has significant deficiency w.r.t. albedo based on Fig. 2. This should be also mentioned.

*Sure, we will mention that; however, this is not specific to our method but a general problem of satellite products.*

Figure 4: I suggest to add legend of point colors into the figure for easier reading.

*Ok.*

Figure 6,7: maybe it's better to add grids in the figure.

*Ok.*

**Citation**: https://doi.org/10.5194/amt-2023-188-RC1
*Thank you for your feedback.*

Review of the manuscript titled "On the production and validation of satellite-based UV index maps" of Schenzinger et al.

The manuscript presents a method for generating high-resolution UV index maps of Europe at 15-minute intervals. The method is tuned to enhance computational efficiency. The work is important as such timely UV index maps disseminate important information about the actual UV level to the general public. However, before publication, the manuscript should be improved taking into account both the general and specific comments.

General comments:

The manuscript's title should be revised to better encapsulate the essence of the research.

*Suggestion: A method to produce satellite based UV index maps and the problems with their validation*

Moreover, the abstract should be rewritten to include the most important results in a more precise way. The plots are good and comprehensive, but please check that all of them are appropriately referenced within the text. Additionally, the methodology should be explained more clearly, please see the specific comments. I think the suggestion of using CMF for validation of satellite UV should already be included in the abstract.

*We will quantify the quality statements and add the statement about the CMF for validation.*

Furthermore, it is essential to elucidate the practical applications of these UV index maps and their significance, particularly with regard to the general public.

*We will add information regarding the establishment of the Austrian UV network and its role of public information to the introduction. Furthermore we will stress the importance of maps for complementing information from ground stations, which are not present everywhere.*

Specific comments:

Abstract:

Please be more specific: ..." data agrees well" -> show numbers
..." gets higher" -> how much higher?

*See general comment above*

Page 1, line 11, I think you should add a sentence or two about the Arctic ozone loss which occurs almost every spring (not only in 2020). And please check the reason for the record low ozone e.g. in Benrhard et al., 2020, and references therein:

Bernhard, G. H., Fioletov, V. E., Grooß, J.-U., Ialongo, I., Johnsen, B., Lakkala, K., Manney, G., Müller, R., Svendby, T. : Record-breaking increases in Arctic solar ultraviolet radiation caused by exceptionally large ozone depletion in 2020. Geophysical Research Letters, 47, e2020GL090844. https://doi.org/10.1029/2020GL090844 , 2020

For Arctic springtime ozone loss in general, see e.g.,

Bernhard G., Fioletov V., Grooss J.-U., Ialongo I, Johnsen B, Lakkala K, Manney G., Müller R., Svendby, T., 2023: Ozone and UV radiation [in "State of the Climate in 2022"], Bull. Amer. Meteor. Soc., 104 (9), S 308 -S 310 , https://doi.org/10.1175/2023BAMSStateoftheClimate.1.

*Yes, we will do so. Thank you for providing the references.*

Page 1, line 20: You could add that those proxy data can be used as input to radiative transfer models to produce satellite UV products, from which the maps can be plotted. And add references to TOMS/OMI/TROPOMI algorithm papers.

*Ok.*

Page 1, line 22: "A lot of previous works distinguish cloud-free from cloudy situations by employing a radiative transfer model for clear sky calculations and a separate one to account for the cloud effects (Verdebout, 2000; Schallhart et al., 2008; Chubarova et al., 2012;

Lakkala et al., 2020)". This is not the case for the TROPOMI algorithm (Lakkala et al., 2020). For TROPOMI, there are two lookup tables, the first calculates cloud optical depth, and the second one directly all sky UV. Clear sky UV is not calculated separately. See e.g.

Lindfors, A. V., Kujanpää, J., Kalakoski, N., Heikkilä, A., Lakkala, K., Mielonen, T., Sneep, M., Krotkov, N. A., Arola, A., and Tamminen, J.: The TROPOMI surface UV algorithm, Atmos. Meas. Tech., 11, 997–1008, https://doi.org/10.5194/amt-11-997-2018, 2018.

*We will remove the reference. Apologies for the misunderstanding.*

Page 2: line 41: ..." especially the ozone concentration, but also aerosol optical depth." -> + pollutants?

*Pollutants other than aerosol are usually negligible when it comes to surface UV radiation; however, we will change the sentence to cause less confusion.*

Page 1 and 2: It's difficult to follow what is new in your study compared to the method of Schallhart et al., 2008. Please reformulate the way you explain an already existing method (Schallhart et al., 2008) and your new method. E.g. in line 49 you write "We only do this calculation..." Do you mean in Schallhart et al., 2008 or is it something new that your study implements?

*We will be more specific. The main difference is that we only use one linear interpolation to calculate the CMF compared to two lookup tables (one for the liquid water content and one for the CMF). Furthermore, we do not find cloudy pixels, but outsource this task to the cloudmask input.*

Page 2, line 55: Specify the inputs you use from CAMS: total ozone ....+ ...+
Do you get the beta parameter also from CAMS or from where?

*Ok. We calculate beta from CAMS AOD; we will add this information accordingly.*

Page 3, lines 69-79: It took me a lot of time to understand what you have done in Figure 2. And still, it's not clear if the plot is only based on model calculation. Is the text related to ground measurements related to Fig. 2? I suppose not.

*See response to Referee 1*

Then I suggest that the place of the ground instrumentation is not in this Section 2.1. If you think they are in the right place, then open up more, about how you use them.

*We will change the section to Data and Methods with one subsection on the UV map production and the other one on the ground measurements with some more details on those.*

Figure 2: Why don't you make a plot of the CMFs? x-axis calculated with scaled 600nm radiance and y-axis with 300nm clear sky/all sky?

*See response to Referee 1*

Figure 2: Please explain the nonlinearity due to higher ground albedo.

*We are not really sure where exactly this is coming from; however, it is not relevant for our method as we do a linear approximation anyway.*

Page 3, line 72: "To be able to compare the results for different albedos.." -> Please open a little bit: Do you mean different cloudiness + ground albedo conditions, which are seen from the satellite as "one combined albedo"?

*We hope this becomes more clear with the change in figure 2, but we will also change the statement to be easier understandable.*

Page 4, line 92: Please include the uncertainty or error range for all r-values you show.

*We mentioned the coefficient of determination r² as a reference for fit quality. However, we will also make sure that the slope of the fit is not named the same as a correlation to avoid confusion.*

Page 4, line 97: "the clear sky model has a positive bias, i.e. the satellite map.." I wonder, if you say that the clear sky model is based on input from CAMS, shouldn't you take directly the CAMS clear sky UV index product? Anyway, the bias is mostly due to the bias in the input of the clear sky calculations.

*We cannot use CAMS clear sky UV due to the insufficient resolution. Furthermore we have the option to adopt different sources for our input parameters if the necessity arises. (E.g. there is a better product available for one of them, there is a problem in CAMS availability, ...)*

Page 5, line 121: "as large differences" -> as large absolute differences (when you look at relative differences it's the other way round).

*We will change this.*

Page 5, Figure 5 is not referred.

*There are two references in the text; lines 105 and 118*

Page 5. I don't really see the point of comparing a satellite retrieval with 30-minute average ground measurements. The ground measurements should be 1-5 min data or 1 scan or so.

*We need to use 30 minute means due to consitency. While most of the Austrian stations provide 10min means (and we would be able to get some in even higher resolution), the smallest common denominator is 30 minutes. On a clear day, the difference between the 30min mean and the measurement point of the satellite is negligible, though it can lead to errors with rapidly changing cloud conditions.*

Figure 8: Why the satellite-model didn't capture the rain at 14 UTC? From the all-sky camera, it seems to have been an overcast situation.

*The satellite does show clouds, just the CMF is not as low as for the ground; however, we will add the time scale to the UV index plots as well in case there was a misattribution.*

Page 6, line 162: "In these particular meteorological conditions..." Please specify which kind of conditions

*We will expand on this; it is mostly summer convective clouds*

Page 6, line 193: "That leaves an erroneous input for aerosol as the source of error" -> What do you mean?

*We will change this statement for less confusion; basically we wanted to say that we can exclude other major error sources, so the most logical conclusion is that the error is due to aerosols.*

Page 7, line 192/199: "Can't you force the albedo to be around 0.05 for snow-free surface?

*It is possible, but we do not see the advantage in doing so (in some cases this would even lead to more problems). Furthermore, in the complex terrain of the Alps, "snow free" is not straightforward to establish and the effective albedo at a point is determined by a broader area than just the local ground albedo.*

Page 7, paragraph starting at line 214: You should specify that you are first talking of the ground albedo-> modeled clear sky UV index. And that in the next sentence you talk about the albedo the satellite sees (600nm). From the reflected radiation, the satellite can't know if it's reflected from clouds or from snow. For the satellite, clouds and snow look similar - a reflecting surface. At least this is the case for OMI/TROPOMI instruments, and they underestimate UVindex over snow as they can interpret snow erroneously as a cloud on a clear day.

e.g.Bernhard, G., Arola, A., Dahlback, A., Fioletov, V., Heikkilä, A., Johnsen, B., Koskela, T., Lakkala, K., Svendby, T., and Tamminen, J.: Comparison of OMI UV observations with ground-based measurements at high northern latitudes, Atmos. Chem. Phys., 15, 7391-7412, doi:10.5194/acp-15-7391-2015, 2015.

or Lakkala et al. 2020 If you mean something else, please rephrase.

*We will rephrase and include the corresponding references.*

Page 8, line 232: "This approach itself is widely employed...". Not in the TROPOMI algorithm (Lakkala et al.,2020).

*We will remove the reference. Apologies for the misunderstanding.*

Data availability: It's not clear if the presented method is already in operational use at https://uv-index.at/map/ ?

*The method is already used in these maps.*

**Citation**: https://doi.org/10.5194/amt-2023-188-RC2
*Thank you for your feedback.*